# MATTERS ARISING

# Pitfalls in using phenanthroline to study the causal relationship between promoter nucleosome acetylation and transcription

Sevil Zencir[1], Daniel Dilg[1], David Shore [1✉] & Benjamin Albert [2✉]

ARISING FROM Martin et al. *Nature Communications* https://doi.org/10.1038/s41467-020-20543-z (2021)

A recent study by Martin et al.[1] explored the cause-and-effect relationship between histone acetylation and transcription and reached the conclusion that transcription is a prerequisite for histone acetyl transferase (HAT) activity at both promoters and gene bodies. This thought-provoking proposal is particularly surprising with respect to gene promoters since it contradicts the widely accepted model in which histone acetylation occurs upstream of, and typically stimulates, RNA polymerase II (RNAPII) recruitment and transcription initiation. The conclusions in this paper are based upon the assumption that 1,10 phenanthroline (1,10-pt from hereon), the primary agent used in this study to inhibit RNAPII, has no effect on the transcription factors (TFs) that recruit HAT and histone deacetylase (HDAC) complexes to promoters. We show here that this is not the case since 1,10-pt treatment rapidly alters major signaling pathways that have profound genome-wide effects on the localization and activity of TFs, as well as HAT and HDAC complexes. The causal relationship between promoter nucleosome acetylation and RNAPII transcription thus cannot be inferred using this experimental approach.

Establishment of histone acetylation patterns at gene promoters probably involves a complex balance between the opposing activities of HAT and HDAC complexes, modulated by condition-dependent binding or activity of TFs, which recruit these complexes to promoters. For example, two major growth-promoting TFs in yeast, Ifh1 and Sfp1, are rapidly displaced from ribosomal protein (RP) and ribosomal biogenesis (RiBi) gene promoters, respectively, following inhibition of TORC1 kinase. This is associated with release of the essential NuA4 HAT complex, the simultaneous recruitment of the Rpd3L HDAC[2–4], and a rapid (<10 min) decrease of both transcription and promoter histone acetylation at these genes[5]. The concomitancy of these events makes it difficult to infer a cause-effect relationship between RNAPII activity and acetylation.

One possible experimental approach towards untangling this relationship would be to rapidly block transcription initiation without affecting the activity of major growth or stress signaling pathways, which are known to modulate HAT and HDAC targeting at promoters. A previous study[6] used a thermosensitive allele of the large subunit of RNAPII (*rpb1-1*) to inhibit mRNA synthesis and concluded that promoter histone acetylation is not affected by transcription arrest. However, Martin et al.[1] noted that subsequent studies raised concerns regarding the efficiency of Rpb1 inactivation at the non-permissive temperature and they thus chose to use the transcription inhibitor 1,10-pt to test the dependence of acetylation on transcription.

The problem with this approach is that 1,10-pt treatment rapidly activates a strong stress response that affects RNAPII promoter recruitment genome-wide, as clearly seen in the Martin et al.[1] Rpb3 ChIP-seq data. For example, peaks of RNAPII binding are induced by 1,10-pt at the promoters of all Heat Shock Factor 1 (Hsf1) target genes (defined in ref. [7]) and at plasma membrane transporter genes that are specifically up-regulated in response to environmental stress or drug treatment (two examples of each are shown in Fig. 1A, B). In contrast, the promoters of many (at least 1500) growth-related genes, including almost all RP, RiBi and RiBi-like genes[8], and representing well over 50% of all RNAPII initiation events in rapidly growing cells, rapidly lose RNAPII promoter binding (see Fig. 1C for examples). However, genes whose expression is less sensitive to stress, such as *ACT1* or *CDC19*, maintain peaks of RNAPII binding at their promoters (Fig. 1D). Consistent with a heat shock-like stress response, gene ontology (GO-term) analysis of the 1,10-pt effect on RNAPII promoter binding resembles that seen following heat shock (Supplementary Data 1) and both H3K23ac and H4K8ac changes at highly expressed genes induced by 1,10-pt strongly correlate with RNAPII binding changes observed upon heat-shock (Fig. 1E).

---

[1] Department of Molecular Biology and Institute of Genetics and Genomics of Geneva (iGE3), 30 quai Ernest-Ansermet, CH 1211 Geneva, Switzerland. [2] Molecular, Cellular & Developmental Biology (MCD), Center for Integrative Biology (CBI), University of Toulouse CNRS/UPS, Bâtiment IBCG, 118, route de Narbonne, 31062 Toulouse, France. ✉email: David.Shore@unige.ch; Benjamin.Albert@univ-tlse3.fr

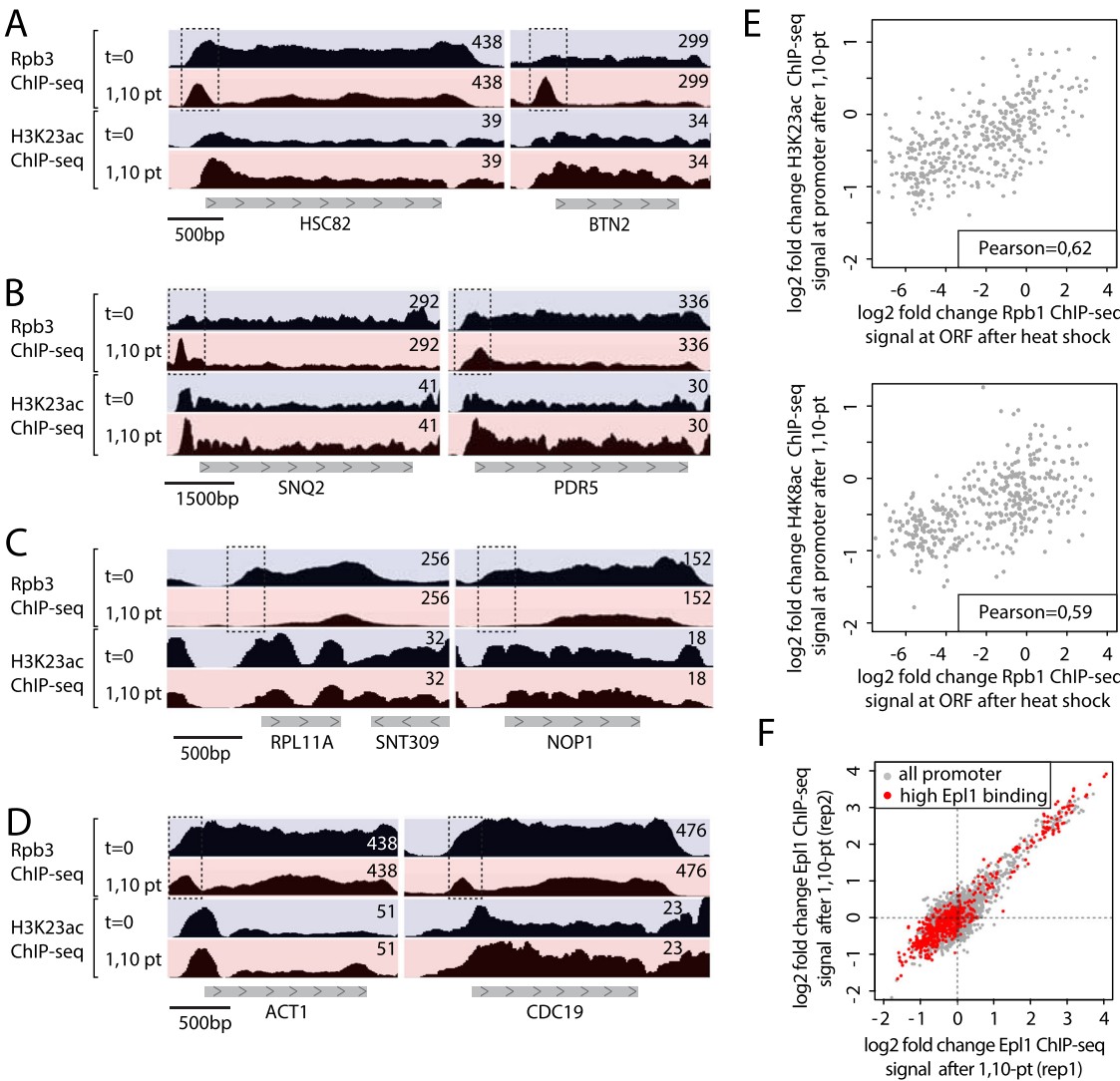

**Fig. 1 1,10-pt induces a stress response affecting RNAPII and Epl1 promoter binding. A–D** Genome browser tracks showing RNAPII (Rpb3) or H3K23 acetylation ChIP-seq read counts at the indicated genes before ($t = 0$) or 15 min after 1,10-pt treatment. Dashed boxes indicate promoter regions where Rpb3 binding either decreases or increases (possibly in an elongation-block state) following 1,10-pt treatment. Raw data are from Martin et al.[1]. **E** Scatter plot comparing H3K23ac (top) and H4K8ac (bottom) at promoters 15 min following 1,10-pt treatment ($y$-axis) to Rpb1 ChIP-seq (all fold-change, log2) at 5 min following heat shock ($x$-axis) for the 400 most highly expressed genes. Pearson coefficient is indicated. Raw data are from Martin et al.[1] for H3K23ac and H4K8ac ChIP-seq and[13] for Rpb1 ChIP-seq. **F** Scatter plot comparing two replicates (Rep1 and Rep2) measuring log2 fold change of Epl1 ChIP-seq signal before and after 1,10-pt treatment at all RNAPII promoters (gray dots) or at 562 promoters displaying high Epl1 binding in the absence of 1,10-pt (red dots), selected by Martin et al.[1]. Raw data are from ref.[1].

None of the above should come as a surprise since two previous studies indicated that 1,10-pt induces a heat shock-like response[9,10] and a more recent study showed that 1,10-pt, even at a dosage 4-fold lower than that used by Martin et al.[1], leads to the complete and rapid inactivation of TORC1, Sch9 and PKC kinases, as well as partial induction of the stress-responsive Hog1 kinase[11]. We speculate that Hsf1 activation may result from a decrease in RNAPI activity[9,12], which unleashes the ribosome assembly stress response (RASTR;[13]). Interestingly, in addition to Hsf1 activation, RASTR causes strong down-regulation of nearly all RP genes[8,14], an effect also observed in the Martin et al.[1] Rpb3 ChIP-seq data.

This loss of RNAPII promoter binding following 1,10-pt treatment is unlikely to be a direct effect of 1,10-pt since the exact opposite is observed at stress-induced genes. Instead, it is best explained by the release of key activator TFs (e.g., Ifh1 and Sfp1) due to TORC1 inhibition by 1,10-pt[11]. Consistent with this view,

Eshelman et al.[11] demonstrated a rapid (<10 min) decrease in nuclear Sfp1 following 1,10-pt addition, concomitant with an increase in the nuclear import of Dot6, which recruits the Rpd3L HDAC to many RiBi and RiBi-like genes[2]. It would therefore appear that 1,10-pt causes widespread changes in activator and repressor binding, which in turn drives a major genome-wide redistribution of HDAC and HAT complexes. Consistent with this notion, Epl1 ChIP-seq analysis of the NuA4/Piccolo HAT complex by Martin et al.[1] reveals important changes following 1,10-pt treatment (Fig. 1F; see below). Consequently, the use of 1,10-pt does not allow one to understand whether the observed acetylation changes are due to a loss of RNAPII activity or changes in HAT and HDAC binding.

We also take issue with the Martin et al.[1] analysis of their Epl1 ChIP-seq data, which they use to argue that 1,10-pt treatment has no significant effect on promoter binding of the essential NuA4 HAT complex. They focus on a set of 562 genes that display peaks

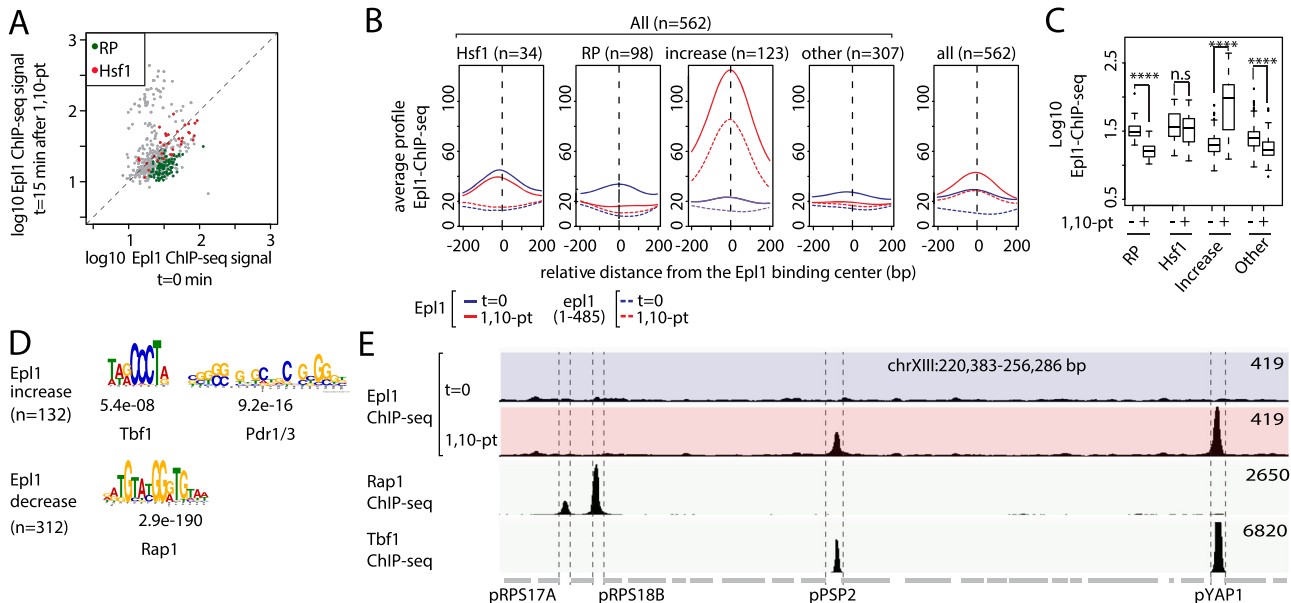

**Fig. 2 1,10-pt treatment mostly leads to a significant decrease in Epl1 promoter binding except for a small group of genes displaying a dramatic Epl1 binding increase. A** Scatter plots comparing Epl1 ChIP-seq signal before and after 1,10-pt treatment at promoters of all 562 genes (gray dots) selected by Martin et al.[1] to study Epl1 promoter binding. Indicated gene categories (RP genes, n = 98; Hsf1 target genes, n = 34) are color-coded on the scatter plots; raw data are from replicate 1 from ref. [1]. **B** Average Epl1 or Epl1(1–485) (dotted line) binding profiles on genes selected by Martin et al.[1]) before (blue curves) or after 15 min of 1,10-pt treatment (red curves). The first four panels show data for all 562 genes divided into four separate categories, as marked: Hsf1 target genes (Hsf1, n = 34), ribosomal protein genes (RP, n = 98), genes where Epl1 ChIP-seq signal increases >1,2 fold (Increase, n = 123), and all other genes in this group (Other, n = 307). The right-most panel shows the average for all 562 genes. Data are taken from Martin et al.[1] and all plots are centered on the Epl1 signal peak. **C** Boxplots of log10 EpL1 ChIP-seq signal at promoters of the four gene categories defined in **B** before (−) or after (+) 15 min of 1,10-pt treatment. The ChIP-seq signal is quantified in a region delimited as the 100 bp region upstream and downstream of the Epl1 signal peak defined by Martin et al.[1]. Asterisks show significant difference according to the Wilcoxon test (*$P < 0.05$, **$P < 0.01$, ***$P < 0.001$, ns not significant). **D** Most significant motifs found in genes at which Epl1 binding increases (>1,2-fold) or decreases (<0,8-fold) after 1,10-pt treatment. E values and the TF associated with the consensus binding motif identified by MEME are indicated. **E** Genome browser tracks showing Epl1, Rap1 and Tbf1 ChIP-seq read counts at the promoter regions (roughly demarcated by dotted lines) of the indicated genes before (t = 0) or 15 min after 1,10-pt treatment. Raw data are from Martin et al.[1], Knight et al.[17], and Preti et al.[18] and for Epl1, Rap1, and Tbf1 ChIP-seq, respectively.

of Epl1 promoter binding and claim, showing only average plots, that binding remains high following 1,10-pt treatment. However, their average plots hide the fact that this group of genes responds in a very heterogenous manner, with a subgroup displaying a dramatic binding increase in response to 1,10-pt treatment (see Figs. 1F and 2A). To clarify the behavior of these 562 genes, we re-analyzed the data by separating out RP (n = 98, all Rap1-bound), Hsf1 target (n = 34), and Epl1 binding increase genes (n = 123) from all other genes (n = 307). This shows that Epl1 ChIP-seq signal significantly decreases at most of these 562 genes following 1,10-pt treatment (over-represented by growth-related genes; see Supplementary Data 2 for GO-term analysis) but instead spikes at a relatively small subgroup that dominate the average plot of all 562 genes (Fig. 2B, C). Epl1 promoter binding clearly decreases at RPs enriched in the Rap1 binding motif (Fig. 2D), contrary to the claim by Martin et al.[1], which was based upon a group average. Our motif analysis (Fig. 2D) further demonstrates that promoters at which Epl1 binding increases following 1,10-pt treatment are enriched in Tbf1 binding sites (examples of Epl1 binding at Tbf1- or Rap1- bound promoters are shown in Fig. 2E) or in the Pdr1/3 binding motif, associated with drug resistance genes such as *SNQ2* and *PDR5* (shown in Fig. 1). Why this subgroup of promoters displays a massive increase in Epl1 binding upon 1,10-pt treatment remains to be addressed. In this regard, we note that the Epl1(1–485) mutant, which is thought to abolish promoter recruitment of NuA4, shows no promoter binding at most genes in either condition, as expected, but nevertheless displays a clear peak in promoter

binding following 1,10-pt treatment at the same small subgroup of outlier genes (Fig. 2B).

Finally, we note that Martin et al.[1] (and in their reply to this letter) contend that if one looks globally at the ~5000 promoters where they report Epl1 binding one finds little of no effect of 1,10-pt treatment. However, Epl1 binding is at background levels at most promoters (the reader is invited to examine the data on a genome browser), and the same is true for other NuA4 subunits mapped by ChIP-exo (see http://yeastepigenome.org/). It is thus not possible to know from these data whether Epl1 (or NuA4) binding is altered by 1,10-pt treatment genome-wide. Nevertheless, at those (562) genes where Epl1 promoter binding is detectable by ChIP-seq above background levels, 1,10-pt treatment mostly leads to a significant decrease in binding (Fig. 2). Furthermore, the claim by Martin & Howe that the difference in our interpretations might result from our inappropriate definition of promoter regions is also invalid since our display of their Epl1 ChIP seq data is centered on the peaks of binding exactly as in their own analysis.

In summary, we argue that the claim made by Martin et al.[1] that promoter-bound HATs are unable to acetylate histones in the absence of transcription is not supported by their data. A rigorous test of this idea will require an experimental strategy that allows one to abolish RNAPII transcription without perturbing the signal transduction pathways or the downstream TFs and histone modifiers (HATs and HDACs) whose binding and activity they modulate. The most promising approaches may be rapid and specific depletion of either RNAPII or the TATA

binding protein (Tbp1), using the anchor-away or auxin-induced degron (AID) methods. Martin et al.[1] have in fact performed an AID experiment targeting the Rpb2 subunit of RNAPII, but they report only a few histone western blots on this strain which, interestingly, show an overall weaker decrease in acetylation compared to 1,10-pt treatment, even though the depletion would appear to be complete. Further characterization of the Rpb2-AID strain, or readily available anchor-away strains targeting Rpb1 or Tbp1, are likely to clarify the causal relationship between histone acetylation at both promoters and gene bodies and RNAPII promoter recruitment and transcription.

## Methods

**Data analysis**. ChIP-seq datasets (Replicate 1) from 1,10-pt-treated or untreated cells were normalized to silent regions by ChIP Normalization Factors calculated in ref. [1]. ChIP-seq signal was quantified for each promoter defined as the 400 bp region upstream of the transcription start site (TSS), as obtained from ref. [15]. To calculate ChIP-seq signals fold change, a ratio between the total number of reads for each promoter from 1,10-pt treated cells and the number of reads from untreated cells was made. RP gene list was defined in ref. [14] and Hsf1 gene list in ref. [7].

**Statistics**. Yeast Mine (https://yeastmine.yeastgenome.org/yeastmine/begin.do) was used to determine GO-terms and MEME[16] was used for motif enrichment analysis, with the following settings: anr -nmotifs 5 -minw 6 -maxw 20 -objfun classic -revcomp -markov_order 0.

In all of the box plots, the box shows the 25th–75th percentile, whiskers show the 10th–90th percentile, and dots show the 5th and 95th percentiles. Asterisks show significant difference according to the Wilcoxon test ($*P < 0.05$, $**P < 0.01$, $***P < 0.001$, ns not significant).

**Reporting summary**. Further information on research design is available in the Nature Research Reporting Summary linked to this article.

## Data availability

Data used in this study are from the NCBI Gene Expression Omnibus under the following accession codes: "GSE110287", RNAPII ChIP-seq ± 1,10-pt, Epl1 and H3K23ac/H4K8ac ChIP-seq ± 1,10-pt; "GSE110286", Rpb3 ChIP-seq; "GSE125226", Serine-5 Rpb1 ChIP-seq ± Heat Shock; "GSE20870", Tbf1 ChIP-seq; "GSE61596", Rap1 ChIP-seq.

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

## Acknowledgements

We thank members of the Shore lab and Fred van Leeuwen for discussions, Nicolas Roggli for help with artwork, and the Swiss National Fund (grant number 31003A_170153 (S.Z., D.D.) and the Republic and Canton of Geneva for financial support (D.S.). B.A. is supported by Grants from ANR JCJC (RASTR) and funding from CNRS and University of Toulouse.

## Author contributions

B.A. and S.Z. initiated this study, and carried out most of the analysis. B.A. wrote the first draft, which was edited by the other authors (D.S., D.D.).

## Competing interests

The authors declare no competing interests.
