## [Peer Review File · Nature Communications]

Pitfalls in using phenanthroline to study the causal relationship between promoter nucleosome acetylation and transcriptionReviewers' Comments:

Reviewer #1:

Remarks to the Author:

The original paper from Martin et al. made the provocative suggestion that the majority of histone acetylation in yeast cells is targeted by active transcription, reversing the usual cause and effect model. Many, but not all, of the experiments involved using the chemical 1,10-phenanthroline to inhibit transcription. Histone acetylation dropped at both promoters and downstream regions upon 1,10-pt treatment, likely by reducing recruitment via RNA pol II, but not disrupting the recruitment of NuA4 HAT (assayed via the Epl1 subunit) to promoters by sequence-specific factors. Other treatments to inhibit transcription had a similar effect, but the analysis of these other conditions was not very extensive.

The correspondence from Zencir et al. questions whether the 1,10-pt effects can be directly and solely assigned to transcription inhibition, or whether other mechanisms might be responsible. They make a convincing case that 1,10-pt triggers a stress response, and use some data from the Martin paper to show that Epl1 binding was in fact changing at a significant fraction of bound promoters. The changes correlate with the particular transcription activator used at the promoters, suggesting that 1,10-pt is affecting specific signaling pathways rather than working through a general effect. Overall, I think this analysis raises important questions and is something that readers of the original Martin paper should take into consideration.

One issue I do have with the Zabnir analysis, which was also raised in the reply from Martin and Howe, is that they analyze only H3K27ac data to make their arguments about general histone acetylation. However, the Martin paper is primarily about NuA4 (Epl1), which is an H4 acetyltransferase. It would seem prudent to analyze the NuA4-linked H4 acetylation data to be sure the stress- and growth-related gene expression changes cited in support of their argument are also reflected in H4 acetylation.

In the reply from Martin and Howe, they concede that Epl1 binding does change at some promoters (their Fig 2a is strikingly similar to Fig 2a in Zabnir), but suggest that isn't sufficient to account for the loss of histone acetylation, particularly because 90% of promoters don't have Epl1 peaks (and therefore show no difference in binding with 1,10-pt), but still show a drop in acetylation (Fig 1a). They argue that the differing interpretations are at least partly due to differing definitions of promoter regions. I'm not convinced that's a major issue, and I note that in responding to my original review, the authors indicated that the majority of acetylation is not occurring at promoters anyway. Overall, I think the reply makes a reasonable defense their interpretation of data from original paper. Readers can consider both points of view and come to their own conclusions about which they think is most likely.

One comment about using the fact that TSA inhibition of HDACs did not diminish the response to 1,10-pt as support for the transcription-dependence model. I didn't pick up on this before, but it appears that deletion of Rpd3 (see Fig S2C), and to some extent Hda1, does significantly abrogate the acetylation drop caused by 1,10-pt transcription inhibition. This would seem to undercut the TSA argument.

Together, the two letters demonstrate how science is supposed to work. Opinions can differ, even when looking at the same data. Let's hope future experiments will weigh in to help the field come to a consensus.

Reviewer #2:

Remarks to the Author:

Having read this Comment on the manuscript published in Nature Communications by Martin et al., and the accompanying response as well as the original manuscript, my comments are as follows:

Zencir et al. provide convincing evidence that treatment with 1,10-phenanthroline ("1,10-pt") causes a stress response that results in induction of many stress response genes and repression of other genes, most notably ribosomal protein genes and ribosome biogenesis genes. This effect was not considered in the manuscript by Martin et al., which is certainly a deficiency. I also agree with Zencir's criticism of the use of line graphs that average data for all genes, or subsets of genes, in the Martin et al. manuscript, as this can obscure effects seen at a limited number of genes or result in heterogeneous behavior averaging out to a net zero effect, among other problems. However, it is not clear to this reviewer that it invalidates their conclusions entirely. In particular, the heat maps of Supplemental Figure 3 show that the large majority of genes show loss of Pol II and accompanying loss of H3K23ac, H4K8ac, and H4K12ac; the number of upregulated genes (showing Pol II gain) appears small, consistent with the relatively small number of genes indicated as being induced by 1,10-pt by Zencir et al. Supplemental Figure 5 also is convincing in showing that Epl1 association appears to decrease only modestly at the large majority of genes after 1,10-pt treatment. It seems likely that the RP genes, which are evidently repressed by 1,10-pt treatment and which have Rap1-bound promoters, are at the bottom of the heat map in Supplementary Figure 5 of Martin et al. Martin et al., in their response, address the issue of how such disparate conclusions were arrived at by the two parties from analyzing the same data. The argument that defining promoters in this instance as the region 400 bp upstream of TSS's, as Zencir et al. have done in their analyses, is inappropriate has some merit. I'm not necessarily convinced that this accounts for all the difference. I also agreed with the general criticism from Zencir et al. that the novel idea (albeit supported by some additional studies) that activity of promoter-bound HATs is regulated post-recruitment deserves rigorous testing, and that experiments using anchor away or rapid degradation to rapidly abrogate transcription would have been useful, particularly since they allow a stronger control than in the 1,10-pt experiments, namely administration of auxin (or rapamycin) to untagged strains, as in Supplemental Figure 2b of Martin et al.

Martin et al. also object to the way in which some of the figures in the Comment by Zencir et al. are presented; I found these to be relatively minor objections as the figures in Zencir in general seemed, if sometimes somewhat idiosyncratic, to support their arguments in a fair manner. The exception to this might be Figure 2D, where a case could be made that the varied scales in the y-axis misrepresent the results. Even there, I am not convinced that rescaling so all results were presented with the same axis would affect interpretation.

Two reservations to the original Martin et al. manuscript that occurred to me that were not expressed by Martin et al. were 1) the possibility of nucleosome turnover affecting the degree of modified histones at genes was not considered and 2) loss of Epl1 at gene bodies, clearly seen, could affect acetylation at regions upstream. With regard to this latter point, the "-1" nucleosome at the region showing essentially no loss of Epl1 exhibited relatively modest decreases in histone acetylation (Supplemental Figure 2 of Martin et al.), and clearly some acetylation was retained (Figure 2f-h), so statements that transcription is required for acetylation of 5' nucleosomes are too strong in the original manuscript.

I fear this review has rambled a bit; my overall feeling is that the objections raised by Zencir et al. deserve publication, as does the response from Martin et al. These are both excellent groups of researchers disagreeing over what to some extent are technical issues, and the discussion will in my view interest a fairly wide group of researchers interested in genomic analyses. The Commentary by Zencir et al. also sheds light on weaknesses in the original Martin et al. paper that temper some of the conclusions drawn there. I can easily envision the manuscript and commentary being a productive topic in many graduate journal clubs.

Minor comments:

1) p. 2 line 32 "the opposing the activities"

2) Figure 1, The dotted boxes, which I believe represent polymerase stalled at the 5' end, need to be defined in the legend.

Reviewer #3:

Remarks to the Author:

Zencir make a convincing and valid point about the potential indirect effect of 1,10-pt, thereby making any of Martin's conclusions drawn from it equivocal. The Martin Reply makes an incorrect assertion about Zencir ("is due to the disruption of ...). Zencir do not argue that the results are due to indirect effects, but only that the possibility cannot be excluded. Zencir et al make a compelling case that 1,10-pt is a broadly acting inducer of stress in yeast. Interpretations of Martin et al experiments are based on the assumption that 1,10-pt is a specific inhibitor of RNAPII. Based on these tenets, we can join the Martin and Zencir claims in that a general stress response inhibits RNAPII, which other studies have shown. It is known that RNAPII inhibition persists for 5-15 min, followed by a gradual reactivation. 1,10-pt was used by Martin for 15 min. The crux of the problem is whether the set of experiments reported by Martin et al can really make a definitive statement about whether histone acetylation is a cause or a consequence of transcription. As Zencir point out 1,10-pt might have indirect effects, including genome-wide reprogramming of transcription and/or the signaling cascade brought on by this stress, leading to loss of acetylation by any number of mechanisms.

Martin et al and their Reply assume that NuA4 (Epl1) binds the vast majority of promoters or gene bodies (thereby justifying the use of 5,542 promoter in their denominator for reporting the fraction of genes losing Epl1 in the presence of 1,10-pt). However, no evidence was presented that shows Epl1 binding to this approximate number of promoters. The assumption of Epl1 detection at most promoters/gene bodies may be statistically no different from background. Therefore, any changes in such background signal (i.e., unsubstantiated Epl1 signal) is meaningless. The statement in Martin's Reply: "the key point is that Epl1 occupancy is largely unaffected by 1,10-pt..." is therefore invalid.

Both Zencir and Martin agree that there is considerable Epl1 loss (on average) where Epl1 binding can be detected. However, Zencir make the point that gene averaging obscures heterogeneous effects where some gene show loss of binding and others show no change or increased occupancy. This is a common and well know problem with gene averaging, and therefore problematic with the Martin study. For example, Fig 1 of Martin shows a gene average of 5206 genes, but as an average it really reflects only those genes that have the most RNAPII to lose (i.e., highly expressed genes) or the most acetylation to lose, and is not actually representative of all genes.

The Martin Reply indicates that even genes that gain Epl1 with 1,10-pt treatment lose the corresponding acetylation. Although the effect is modest, as shown in Fig. 2 of the Reply and in prior publications, the result is consistent with Martin's conclusion that HATs can be recruited to at least some promoters without them being active for acetylation. However, the potential for indirect effects does not allow an unequivocal claim. The results are consistent with their model, but is not unequivocal.

The Martin Reply indicates that Zencir's use of a 400 bp promoter window is flawed. However, the complex being examined here (NuA4) binds to promoters with large NDRs, including RP and other induced genes. This is clearly shown elsewhere when examining other NuA4 subunits (see yeastepigenome.org). As long as these genes are the focus of the analysis, 400 bp is appropriate. Most other genes have small promoters (NFRs), but there is no evidence of Epl1 binding there.

The Martin Reply states that Zencir Fig. 2D is plotted to locally expand the y-axis that enhances the apparent loss of Epl1. The Martin Reply is correct. All y-axes should start at zero. Further, the analyses in both studies generally suffer from a lack of variance reporting in the metaplots.

As a final note, transcription is a cycle. So, inhibition of one part of the cycle (Pol II trxn) results in loss of histone acetylation. As a cycle, loss of histone acetylation could be temporally downstream of Pol II transcription, but also upstream the next transcription cycle. But even if 1,10-pt were acting directly and exclusively on RNAPII, and causation of histone acetylation by transcription is established, I don't think this tells us anything definitive about the role of histone acetylation. Transcription might cause histone acetylation and this acetylation might create a chromatin environment that promotes the next round of transcription. So it is a chicken-and-egg question applied to the transcription cycle that is not resolved here.

Reviewer #4:

Remarks to the Author:

Title: Pitfalls in the use of phenanthroline to study the causal relationship between promoter nucleosome acetylation and transcription

1,10 phenanthroline (1,10-pt) is one of chemical reagent that is known to inhibit RNA pol II transcription. Using this agent, Martin et al. studied the relationship between RNA pol II transcription and histone acetylation and found that although transcription was off upon treatment of 1,10-pt, no change in HAT binding to promoter was observed. In addition, there was no strong correlation between HAT occupancy and histone acetylation levels. Based on this, Martin et al. proposed that histone acetylation by HAT (H4 acetylation by NuA4) requires RNA pol II transcription.

Since this contradicts the previously established model in which histone acetylation is required for RNA pol II binding and transcription initiation, the authors of this manuscript raised concerns on using 1,10-pt to study the relationship between RNA pol II transcription and histone acetylation. The authors emphasized that 1,10-pt can strongly induce expression of stress response genes differentially affecting RNA pol II binding genome-wide. In addition, the authors also argue that 1,10-pt treatment results in a genome-wide change of TF binding enhancing redistribution of HATs and HDACs.

Overall, although it is clear that 1,10-pt treatment has an effect on activation of a small number of stress response genes, I don't think the data presented in this manuscript is enough to claim that the data by Martin et al. do not support their conclusion.

Specific comments

1. In figure 1, the authors showed Rpb3 occupancy and H3K23 acetylation. As NuA4 predominantly acetylates H4 and H3 and H4 acetylation show differential sensitivities to 1,10-pt treatment (Martin et al.), the authors should present H4 acetylation patterns. In addition, it would be better to show the correlation between Log2 fold change Epl1 ChIP-seq signal and Rpb3 ChIP-seq signal after 1,10-pt treatment.

2. Manipulation of Y axes scales in Figure 2D to emphasize the difference in Epl1 binding is not appropriate and statistical analysis should be included.

Reviewer #5:

Remarks to the Author:

In the Matters Arising, Albert and colleagues raise two important issues that argue against key conclusions from Martin et al. 2021, studying the cause-and-effect relationship between transcription and histone acetylation. The points raised by Albert are:

1. Martin et al. used the drug 1,10 phenanthroline which inhibits RNA Pol II transcription, to show that 15 min of treatment to yeast cells results in the loss of histone acetylation from gene promoters and gene bodies, as well as having effects on the binding of enzymes (HATs) that catalyze histone

acetylation. However, Albert raises the point that acute treatment of 1,10 phenanthroline has been shown to affect signaling pathways, transcription factor binding, and induces a heat shock-like stress response. Therefore, the loss of histone acetylation is unlikely to be direct consequence of transcription inhibition by 1,10 phenanthroline.

2. Martin et al. used ChIP-seq to show that 1,10 phenanthroline results in an increase in the binding of a histone H4-specific HAT component Epl1 at a subset of gene promoters, despite the loss of histone acetylation. Based on this, they argued that promoter-bound HATs are unable to acetylate histones in the absence of transcription. Albert and colleagues reanalyzed the Epl1 ChIP-seq data to show that increased binding is limited to a smaller subset of promoters, while most of the promoters analyzed by Martin et al. actually show a loss of Epl1 binding. Mostly based on this analysis, Albert and colleagues argue that loss of histone acetylation upon 1,10 phenanthroline is rather due to lack of HAT binding or recruitment.

In the rebuttal to the Matter Arising, Martin and Howe argue that careful analysis of the ChIP-seq data with strictly (and more appropriately) defined promoter boundaries show both increase and decrease in Epl1 binding upon 1,10 phenanthroline, while histone acetylation is mainly decreased in both cases. They therefore conclude that loss of acetylation cannot be simply explained by loss of HAT recruitment. Martin and Howe's response to this issue is convincing, and highlights the importance of carefully defining genomic regions for data analysis.

However, this argument over HAT binding is restricted to about 1/10th of all transcriptionally active yeast gene promoters at which Epl1 ChIP-seq shows factor binding in absence of the drug; and is therefore a minor/secondary point. Regardless of Epl1 binding by ChIP-seq, 1,10 phenanthroline results in decreased histone H4 acetylation at promoter-proximal +1 nucleosomes at the majority of yeast genes (not just a subset), as highlighted in the rebuttal. This suggests that Epl1 ChIP-seq cannot be treated as a reliable metric for HAT recruitment or activity. Nevertheless, Martin and Howe focused on mainly addressing this issue about HAT recruitment in their response.

The more important question raised by Albert and colleagues is whether the loss of histone acetylation upon 1,10 phenanthroline is a direct consequence of transcription inhibition. They point out that this can be better addressed with acute depletion of RNAPolII using an anchor-away or a degron-based approach. Martin et al. have used a RNAPolII subunit Rpb2 degron in their original study, but limited their analysis to bulk measurements of histone acetylation levels by immunoblotting. Therefore, the genome-wide effects of directly depleting/directly inhibiting RNAPolII on histone acetylation remains unknown. Martin and Howe point to a more recent study showing similar results in human cells using the TFIIF inhibitor Triptolide (which again does not directly inhibit RNA PolII). The rebuttal however demands a more compelling argument in support of a causal role of transcription/RNAPolII for nucleosomal histone acetylation.

In summary, Albert and colleagues have raised very important points to question some key conclusions from a seminal study which addresses a fundamental knowledge-gap in gene regulation. I recommend that this exchange is published with minor revision of the Martin and Howe response to better address the question of causality.

Response to reviewers' comments - NCOMMS-21-30437A

Reviewer #1

"...They make a convincing case that 1,10-pt triggers a stress response and use some data from the Martin paper to show that Epl1 binding was in fact changing at a significant fraction of bound promoters. The changes correlate with the particular transcription activator used at the promoters, suggesting that 1,10-pt is affecting specific signaling pathways rather than working through a general effect. Overall, I think this analysis raises important questions and is something that readers of the original Martin paper should take into consideration."

We thank the reviewer for this positive comment. We would simply add here that the stress response induced by 1,10-pt that we note in the Martin et al. RNAPII ChIP-seq data has been amply demonstrated at the level of signaling pathways (e.g., TORC1 and PKA), particularly in a recent and very detailed study by Eshleman et al. (2020) that we cite in our letter.

One issue I do have with the Zabnir (*sic*) analysis, which was also raised in the reply from Martin and Howe, is that they analyze only H3K27ac data to make their arguments about general histone acetylation. However, the Martin paper is primarily about NuA4 (Epl1), which is an H4 acetyltransferase. It would seem prudent to analyze the NuA4-linked H4 acetylation data to be sure the stress- and growth-related gene expression changes cited in support of their argument are also reflected in H4 acetylation.

This is a good point, and we now include analysis of the H4 acetylation data that show a similar pattern, which does indeed support our argument that 1,10-pt induces a stress response that is well known to alter HAT and HDAC recruitment genome-wide. We thank the reviewer for pointing this out.

In the reply from Martin and Howe, they concede that Epl1 binding does change at some promoters (their Fig 2a is strikingly similar to Fig 2a in Zabnir) but suggest that isn't sufficient to account for the loss of histone acetylation, particularly because 90% of promoters don't have Epl1 peaks (and therefore show no difference in binding with 1,10-pt), but still show a drop in acetylation (Fig 1a). They argue that the differing interpretations are at least partly due to differing definitions of promoter regions. I'm not convinced that's a major issue, and I note that in responding to my original review, the authors indicated that the majority of acetylation is not occurring at promoters anyway. Overall, I think the reply makes a reasonable defense their interpretation of data from original paper. Readers can consider both points of view and come to their own conclusions about which they think is most likely.

To begin with the 2nd issue, our promoter definition (TSS to -400) captures the distribution of NuA4 promoter binding events extremely well, as seen in the recent high-resolution ChIP-exo data from Frank Pugh's lab for 4 different subunits (<http://yeastepigenome.org/>). Moreover, Martin et al. note that the average size of NDRs with Epl1 peaks is wider (mean of 297 vs. 164 bp) suggesting that our choice of a 400 bp window for this specific group of genes is perfectly appropriate. Lastly, in our re-analysis of their Epl1 ChIP-seq data (Figure 2) we center reads over the peak of Epl1 binding, as in Martin et al. (see their Figure 2d, e) and thus consider exactly the same NDR window. Their argument regarding promoter region definition is thus invalid and cannot explain our divergent conclusions. The correct explanation is that we separated out genes that behave in opposing directions to 1,10-pt, and thus showed that Epl1 is significantly decreased at most of these genes, whereas they simply averaged over all 562 genes in the group, which highlights a small sub-group of outliers that display an unexplained spike in Epl1 binding. Martin & Howe have still not acknowledged this fact.

Furthermore, Martin et al. actually do report Epl1 binding by ChIP at most promoters (>5000) and claim that this binding does not change following 1,10-pt treatment. However, examination of the Martin et al. Epl1 ChIP-seq data on a genome browser clearly suggests that Epl1 signal is at background levels at most promoters. This is confirmed by the extensive ChIP-exo analysis referred to above (<http://yeastepigenome.org/>) which showed binding of some NuA4 subunits (e.g., Esa1, Eaf1, Eaf5 and Eaf6; Epl1 was not reported) at the promoters of only 250 to 1100 genes (depending upon the subunit

tested). The bottom line is that neither ChIP-seq nor ChIP-exo can detect NuA4 binding at most promoters (even though rapid Esa1 nuclear depletion followed by H4ac ChIP-seq clearly indicates NuA4 action at nearly all gene promoters; see Bruzzone et al., *Genes Dev*, 2018). Thus, one simply cannot know, at this point, whether Epl1 binding changes following 1,10-pt treatment at this large group of genes. Nevertheless, where Epl1 promoter binding is detectable by ChIP, it is quite clear that its binding is strongly affected by 1,10-pt treatment at most of these sites (our Figure 2). Besides, the analysis of Martin & Howe in their reply letter (Figure 2a) clearly indicates that at least 40% of promoters with high Epl1 signal either increase or decrease following 1,10-pt. So, it appears inconsistent to persist in the claim that Epl1 binding is not sensitive to 1,10-pt. The data show that this is simply not true.

One comment about using the fact that TSA inhibition of HDACs did not diminish the response to 1,10-pt as support for the transcription-dependence model. I didn't pick up on this before, but it appears that deletion of Rpd3 (see Fig S2C), and to some extent Hda1, does significantly abrogate the acetylation drop caused by 1,10-pt transcription inhibition. This would seem to undercut the TSA argument.

We thank the reviewer for pointing this out. He/she is correct: the data shown in Figure S2 in Martin et al. do reveal a significant drop in acetylation either following drug treatment or in mutant cells. Furthermore, the experiment involving TSA treatment following administration of 1,10-pt does not show that the effect of 1,10-pt is independent of HDACs. This is because one can easily imagine that TFs and HATs will fail to return to promoters in the constant presence of 1,10-pt, particularly at the 15-minute time point that they use in their assays. This claim is supported by the findings of Poramba-Liyanage et al. (*Genome Research*, 2020), which we discuss in more detail below (pp. 3-4).

Reviewer #2

Zencir et al. provide convincing evidence that treatment with 1,10-phenanthroline ("1,10-pt") causes a stress response that results in induction of many stress response genes and repression of other genes, most notably ribosomal protein genes and ribosome biogenesis genes. This effect was not considered in the manuscript by Martin et al., which is certainly a deficiency. I also agree with Zencir's criticism of the use of line graphs that average data for all genes, or subsets of genes, in the Martin et al. manuscript, as this can obscure effects seen at a limited number of genes or result in heterogeneous behavior averaging out to a net zero effect, among other problems.

We thank the reviewer for these positive comments on our letter.

However, it is not clear to this reviewer that it invalidates their conclusions entirely. In particular, the heat maps of Supplemental Figure 3 show that the large majority of genes show loss of Pol II and accompanying loss of H3K23ac, H4K8ac, and H4K12ac; the number of upregulated genes (showing Pol II gain) appears small, consistent with the relatively small number of genes indicated as being induced by 1,10-pt by Zencir et al. Supplemental Figure 5 also is convincing in showing that Epl1 association appears to decrease only modestly at the large majority of genes after 1,10-pt treatment. It seems likely that the RP genes, which are evidently repressed by 1,10-pt treatment and which have Rap1-bound promoters, are at the bottom of the heat map in Supplementary Figure 5 of Martin et al.

I think that the reviewer has misconstrued (or forgotten) our main point here, namely that the experiments presented by Martin et al. do not allow one to determine whether promoter nucleosome acetylation requires RNAPII action. This is because 1,10-pt inactivates key growth signaling pathways (e.g., TORC1 kinase) controlling TFs that recruit HATs and HDACs. The fact that RNAPII action and histone acetylation are correlated is not at issue. We specifically challenge the claim that the reported data can be used to argue that transcription causes and is required for promoter histone acetylation.

Supplemental Figure 5 could appear convincing, but if the reviewer were to visualize the genome browser tracks of Epl1 ChIP-seq he/she would quickly realize that there is no signal of Epl1 at most promoters (see above for discussion of other NuA4 components). In any event, this figure, even if one

were to believe the statistical significance of the differences, simply shows that Rpb3 and Epl1 binding changes upon 1,10-pt treatment are correlated. For the reasons stated in our Matters Arising letter and elaborated upon (endlessly) here, this does not establish a causal relationship. Finally, we would note that there is no statistical evaluation given for the heat maps in Supplemental Figure 5.

Martin et al., in their response, address the issue of how such disparate conclusions were arrived at by the two parties from analyzing the same data. The argument that defining promoters in this instance as the region 400 bp upstream of TSS's, as Zencir et al. have done in their analyses, is inappropriate has some merit. I'm not necessarily convinced that this accounts for all the difference.

As we discussed above, our definition of the promoter region matches extremely well with the mapped sites of NuA4 subunit binding (both for Epl1 reported by Martin et al. and the four other subunits by ChIP-exo analysis in <http://yeastepigenome.org/>). Furthermore, as stated above, in our re-analysis of the Martin et al. Epl1 ChIP-seq data (our Figure 2), we centered the reads over the peak of Epl1 binding, exactly as was done in Marin et al. (Figure 2d, e). Their argument regarding promoter definition is thus invalid. Just to repeat: the correct explanation for the “disparate conclusions” on this point (the effect of 1,10-pt on Epl1 binding) is that we separated out genes that behave in opposing directions, whereas they simply averaged over all 562 genes in the group, which, as we clearly showed, highlights a small subgroup of outliers and masks the opposite behavior of the majority of genes where Epl1 binding can be reliably detected. Martin & Howe have still not acknowledged this fact.

I also agreed with the general criticism from Zencir et al. that the novel idea (albeit supported by some additional studies) that activity of promoter-bound HATs is regulated post-recruitment deserves rigorous testing, and that experiments using anchor away or rapid degradation to rapidly abrogate transcription would have been useful, particularly since they allow a stronger control than in the 1,10-pt experiments, namely administration of auxin (or rapamycin) to untagged strains, as in Supplemental Figure 2b of Martin et al.

The additional studies cited by Martin & Howe do not support their model (see below), but the central question here is whether the Martin et al. data support their model. As we argue, based upon the well-documented collateral effects of 1,10-pt on signaling pathways affecting transcription initiation and HAT/HDAC recruitment, and clearly demonstrated in the Rpb3 ChIP-seq data of Martin et al., they do not.

In their reply letter, Martin & Howe refer to a recent study in yeast (Poramba-Liyanage et al., *Genome Research*, 2020) to support their claim that “the binding of transcription factors to most promoters is undisturbed upon 1,10-pt treatment.” We examined this paper and discovered that it shows the exact opposite. This study employs a high throughput TAP-tag / ChIP system, together with a bar-coded reporter gene based on the *TEF1* promoter and terminator (called “Epi-Decoder”; Korthout et al. *PLoS Biology* 2018), to analyze chromatin changes in response to 1,10-pt and another drug. It turns out that the *TEF1* promoter uses Rap1 as its “pioneer TF”, whose binding is stable upon any stress so far examined. The Epi-Decoder system robustly detects Rap1 binding, which, as expected, is unaffected by 1,10-pt treatment. However, the *TEF1* promoter is also bound by the stress-sensitive TF Ifh1 (see Cai et al. *Cell Reports* 2013), which binds to and activates virtually all RP genes, as we noted in our letter (see also Zencir et al. *NAR*, 2020). Strikingly, Ifh1 is detected at the *TEF1* promoter before 1,10-pt treatment but disappears rapidly (by 5 minutes) following

treatment with the drug (see the figure to the right, which displays log₂ of normalized [IP/input] values, with bars indicating SD, as a function of time after 1,10-pt treatment; data taken from Supplemental Table S2 in Poramba-Liyanage et al. 2020). Of further note, Poramba-Liyanage et al. also detected a significant increase in binding of the Rpd3L (HDAC) component Sin3 following 1,10-pt treatment, consistent with our claim that growth-related genes like *TEF1* undergo a rapid reconfiguration in HAT and HDAC complex binding upon 1,10-pt exposure for Ifh1-TAP and Sin3-TAP (see figure). We thus thank Martin and Howe for alerting us to this interesting paper, which perfectly highlights our point regarding the pitfalls of 1,10-pt use in determining the causal relationship between transcription and promoter nucleosome acetylation.

Unfortunately, Epi-Decoder has so far been unable to reliably detect HAT complex binding at the *TEF1* reporter gene, so a more complete picture of HAT/HDAC changes there is still lacking. Furthermore, the *TEF1* promoter is in fact more strongly activated by Sfp1 than by Ifh1 (Albert et al. *Genes Dev* 2019), but this is also missed by Epi-Decoder since Sfp1 binding at *TEF1* (and all RiBi and RiBi-like genes) is essentially undetectable by ChIP, though robustly scored by ChEC (Albert et al. *Genes Dev* 2019). We would remind this reviewer that Eshleman et al. (2020) found that 1,10-pt, at a dosage 4-fold lower than that used by Martin et al. (*Nat Comm* 2020), significantly reduced Sfp1 nuclear levels and caused a dramatic increase in Dot6 localization (Dot6 is a co-repressor that recruits the Rpd3L HDAC complex).

In summary, the Martin & Howe claim that “the binding of transcription factors to most promoters is undisturbed upon 1,10-pt treatment” is not supported by Poramba-Liyanage et al., who in any event looked at only one promoter. Martin & Howe fail to explain the evidence that TF binding at “most promoters” is unchanged by 1,10-pt treatment. In fact, one (of the limited number) of transcription factors that Poramba-Liyanage et al. (2020) were able to reliably detect at the *TEF1* promoter, Ifh1, is indeed rapidly released from the promoter upon 1,10-pt treatment.

The other two studies cited by Martin & Howe as supporting their model were carried out in human tissue culture cells. The first (Zhang et al. *Sci Adv* 7, 2021) reports only on bulk H3K27ac and H3K27me3 by immunofluorescence in whole cells and is thus irrelevant to an argument about promoter nucleosome acetylation. The second (Wang et al. doi:10.21203/rs.3.rs-149042/v1) uses a highly toxic drug (Triptolide) with many putative targets (RNAPII not included). The same concerns that we raise regarding 1,10-pt in yeast would seem to apply here. In summary, none of the studies cited by Martin & Howe can be reasonably construed to support their model. Indeed, the most relevant study (Poramba-Liyanage et al., 2020) clearly does just the opposite.

Martin et al. also object to the way in which some of the figures in the Comment by Zencir et al. are presented; I found these to be relatively minor objections as the figures in Zencir in general seemed, if sometimes somewhat idiosyncratic, to support their arguments in a fair manner. The exception to this might be Figure 2D, where a case could be made that the varied scales in the y-axis mis-represent the results. Even there, I am not convinced that rescaling so all results were presented with the same axis would affect interpretation.

We have now standardized the y-axes in Figure 2B (in revised manuscript). This has no effect on our conclusions, and we would argue that the original presentation makes it easier to gauge the relative fold-change induced by 1,10-pt. Importantly, we also add a boxplot of these data (new Figure 2C) showing variance and the median in the different subgroups, allowing us to demonstrate unambiguously that Epl1 binding decreases significantly (at most promoters where it can be reliably measured) following 1,10-pt treatment. The latter effect is clearly hidden in Martin et al. due to the fact that they showed an average (and not median) of highly heterogenous values.

We also add in Figure 2B the ChIP-seq signal of the epl1-(1-485) mutant for the different subgroups of promoters assessed by Martin et al. Epl1-485 is a C-terminal truncation mutation that abolishes NuA4 formation and promoter targeting but maintains the non-targeted H4/H2A HAT activity of PiccoloNuA4 responsible for global acetylation. As expected, our analysis showed that this mutant is not present at any promoter before 1,10-pt treatment but, surprisingly, its binding increases dramatically at the same specific subgroup of promoters where wild-type Epl1 binding increases. This strongly suggests that the increase observed for Epl1 binding following 1,10-pt treatment occurs through an unknown process independent of the C-terminal tail normally required for promoter targeting of Epl1. It appears

surprising that Martin et al. didn't note this point, but instead use this strong increase at a subgroup of promoters to hide the global decrease of Epl1 binding upon 1,10-pt treatment.

Two reservations to the original Martin et al. manuscript that occurred to me that were not expressed by Martin et al. were 1) the possibility of nucleosome turnover affecting the degree of modified histones at genes was not considered and 2) loss of Epl1 at gene bodies, clearly seen, could affect acetylation at regions upstream. With regard to this latter point, the "-1" nucleosome at the region showing essentially no loss of Epl1 exhibited relatively modest decreases in histone acetylation (Supplemental Figure 2 of Martin et al.), and clearly some acetylation was retained (Figure 2f-h), so statements that transcription is required for acetylation of 5' nucleosomes are too strong in the original manuscript.

We agree but have chosen in our letter to emphasize what we feel are more serious methodological problems with the Martin et al. study that prevent them from determining the causal relationship between promoter nucleosome acetylation and transcription initiation.

Minor comments:

1) p. 2 line 32 "the opposing the activities"

2) Figure 1, The dotted boxes, which I believe represent polymerase stalled at the 5' end, need to be defined in the legend.

We thank the reviewer for pointing out these minor errors, which have now been corrected.

Reviewer #3

Zencir make a convincing and valid point about the potential indirect effect of 1,10-pt, thereby making any of Martin's conclusions drawn from it equivocal. The Martin Reply makes an incorrect assertion about Zencir ("is due to the disruption of ...). Zencir do not argue that the results are due to indirect effects, but only that the possibility cannot be excluded. Zencir et al make a compelling case that 1,10-pt is a broadly acting inducer of stress in yeast. Interpretations of Martin et al experiments are based on the assumption that 1,10-pt is a specific inhibitor of RNAPII. Based on these tenets, we can join the Martin and Zencir claims in that a general stress response inhibits RNAPII, which other studies have shown. It is known that RNAPII inhibition persists for 5-15 min, followed by a gradual reactivation. 1,10-pt was used by Martin for 15 min. The crux of the problem is whether the set of experiments reported by Martin et al can really make a definitive statement about whether histone acetylation is a cause or a consequence of transcription. As Zencir point out 1,10-pt might have indirect effects, including genome-wide reprogramming of transcription and/or the signaling cascade brought on by this stress, leading to loss of acetylation by any number of mechanisms.

We thank the reviewer for this clear summary of our central argument. We would emphasize, though, that our point is not hypothetical ("1,10-pt might have indirect effects"), but rather very clearly demonstrated by the Martin et al. Rpb3 ChIP-seq data and perfectly consistent with the effects of 1,10-pt on TORC1 and other signaling pathways reported in a key "other study" alluded to by this reviewer (Eshleman et al. (2020).

Martin et al and their Reply assume that NuA4 (Epl1) binds the vast majority of promoters or gene bodies (thereby justifying the use of 5,542 promoter in their denominator for reporting the fraction of genes losing Epl1 in the presence of 1,10-pt). However, no evidence was presented that shows Epl1 binding to this approximate number of promoters. The assumption of Epl1 detection at most promoters/gene bodies may be statistically no different from background. Therefore, any changes in such background signal (i.e., unsubstantiated Epl1 signal) is meaningless. The statement in Martin's Reply: "the key point is that Epl1 occupancy is largely unaffected by 1,10-pt..." is therefore invalid.

We thank the reviewer for pointing out this problem with the Martin & Howe analysis of Epl1 binding. In fact, neither ChIP-seq nor ChIP-exo can detect widespread NuA4 binding (see above), thus invalidating the Martin & Howe claims, as this reviewer points out. We now emphasize this point in our revised letter.

Both Zencir and Martin agree that there is considerable Epl1 loss (on average) where Epl1 binding can be detected. However, Zencir make the point that gene averaging obscures heterogeneous effects where some gene show loss of binding and others show no change or increased occupancy. This is a common and well know problem with gene averaging, and therefore problematic with the Martin study. For example, Fig 1 of Martin shows a gene average of 5206 genes, but as an average it really reflects only those genes that have the most RNAPII to lose (i.e., highly expressed genes) or the most acetylation to lose, and is not actually representative of all genes.

The Martin Reply indicates that even genes that gain Epl1 with 1,10-pt treatment lose the corresponding acetylation. Although the effect is modest, as shown in Fig. 2 of the Reply and in prior publications, the result is consistent with Martin's conclusion that HATs can be recruited to at least some promoters without them being active for acetylation. However, the potential for indirect effects does not allow an unequivocal claim. The results are consistent with their model, but is not unequivocal.

We thank the reviewer for highlighting the general problem of gene averaging. Regarding the genes where Epl1 binding increases dramatically upon 1,10-pt treatment (2nd paragraph above), we agree with the reviewer that the corresponding decrease in acetylation is indeed quite modest. Furthermore, most of these genes have high H4K8 and H4K12 Ac levels in both conditions, so inhibition of RNAPII by 1,10-pt has not abolished acetylation, as the Martin et al. model would suggest. A more interesting question, in our opinion, is: why does this relatively small and specific group of genes display such a dramatic spike in Epl1 (and Epl1-485) binding upon treatment with 1,10-pt, a drug that supposedly just inhibits RNAPII? To use the strange and unexplained behavior of these genes in response to 1,10-pt treatment as an argument that transcription is required for promoter acetylation is inappropriate, to say the least. Moreover, the fact that Epl1 can be recruited at these genes independently of its C-terminus, known to be required for TF-mediated promoter binding, just increases the mystery regarding these promoters.

The Martin Reply indicates that Zencir's use of a 400 bp promoter window is flawed. However, the complex being examined here (NuA4) binds to promoters with large NDRs, including RP and other induced genes. This is clearly shown elsewhere when examining other NuA4 subunits (see <http://yeastepigenome.org/>). As long as these genes are the focus of the analysis, 400 bp is appropriate. Most other genes have small promoters (NFRs), but there is no evidence of Epl1 binding there.

We thank the reviewer for pointing this out. Indeed, all the NuA4 components that could be mapped by ChIP-exo at promoters (Esa1, Eaf1, Eaf5 and Eaf6) displayed peaks that are centered on the TSS to -400 bp window that we chose as our promoter definition, as discussed in more detail above. The same is true for the genes with clear Epl1 peaks, which we re-analyze in exactly the same way (centering reads on the Epl1 peaks) as Martin et al. do.

The Martin Reply states that Zencir Fig. 2D is plotted to locally expand the y-axis that enhances the apparent loss of Epl1. The Martin Reply is correct. All y-axes should start at zero. Further, the analyses in both studies generally suffer from a lack of variance reporting in the metaplots.

As stated above, we presented the plots in this form to make the fold-change upon 1,10-pt treatment as clear as possible and to present data in a similar way to Martin et al., thus helping the reader to compare our two analyses. We have now standardized the y-axes, which does not change our conclusions at all, but does have the benefit of emphasizing the bizarre behavior of the group of genes that display a huge spike of (apparent) NuA4 (Epl1) binding upon 1,10-pt treatment, for no known reason. We also add boxplots of the same data to compare more rigorously the different subgroups by defining medians and assigning p-values to the changes observed following 1,10-pt treatment.

As a final note, transcription is a cycle. So, inhibition of one part of the cycle (Pol II trxn) results in loss of histone acetylation. As a cycle, loss of histone acetylation could be temporally downstream of Pol II transcription, but also upstream the next transcription cycle. But even if 1,10-pt were acting directly and exclusively on RNAPII, and causation of histone acetylation by transcription is established, I don't think this tells us anything definitive about the role of histone acetylation. Transcription might cause histone acetylation and this acetylation might create a chromatin environment that promotes the next round of transcription. So it is a chicken-and-egg question applied to the transcription cycle that is not resolved here.

This is an interesting speculation but goes well beyond our aim of pointing out what we see as clear flaws in the Martin et al. analysis. Unfortunately, we do not have the space to discuss this and other related issues in our letter.

Reviewer #4

... the authors of this manuscript raised concerns on using 1,10-pt to study the relationship between RNA pol II transcription and histone acetylation. The authors emphasized that 1,10-pt can strongly induce expression of stress response genes differentially affecting RNA pol II binding genome-wide. In addition, the authors also argue that 1,10-pt treatment results in a genome-wide change of TF binding enhancing redistribution of HATs and HDACs.

Overall, although it is clear that 1,10-pt treatment has an effect on activation of a small number of stress response genes, I don't think the data presented in this manuscript is enough to claim that the data by Martin et al. do not support their conclusion.

In fact, we show clearly that 1,10-pt has much more than "...an effect on activation of a small number of stress response genes", since it also leads to the downregulation of a vast number of growth-related genes (over 500 RP, RiBi and RiBi-like genes, plus ~1000 other genes), consistent with its documented effect on TORC1 and PKA signaling pathways. TORC1 inhibition is well known to affect TF binding, as well as HAT and HDAC recruitment, at many promoters. We thus fail to understand why this reviewer would seem to ignore these points and why he/she thinks that the analysis and arguments we present fail to show that the Martin et al. conclusions are not supported by their data. In all due respect, we feel that this reviewer's opinion is lacking any reasoned justification, and is thus unhelpful, at best.

Specific comments

1. In figure 1, the authors showed Rpb3 occupancy and H3K23 acetylation. As NuA4 predominantly acetylates H4 and H3 and H4 acetylation show differential sensitivities to 1,10-pt treatment (Martin et al.), the authors should present H4 acetylation patterns.

H4K8ac data are now shown in Figure 1E together with the H3K23ac data. The two correlate very similarly with the RNAPII ChIP-seq data following heat shock. We thank the reviewer for pointing this out (i.e., the focus of Martin et al. on NuA4 and thus the higher relevance of H4ac compared to H3ac).

In addition, it would be better to show the correlation between Log2 fold change Epl1 ChIP-seq signal and Rpb3 ChIP-seq signal after 1,10-pt treatment.

Although this is an interesting question, a proper quantitative analysis would be complex and beyond the scope of our Matters Arising letter, which already exceeds the space limitation. Qualitative analysis (see genome browser tracks in Figure 1A-D) already reveals the complex effect of 1,10-pt on Rpb3 distributions, with some genes (e.g., Hsf1 targets and two of the Tbf1-associated genes where Epl1 levels spike) showing induced promoter peaks of Rpb3, in contrast to other genes (e.g., RPGs) where promoter-associated Rpb3 is diminished or absent. These trends are in line with the changes seen at those promoters where Epl1 ChIP-seq signals that can be reliably detected (see our new Figure 2B, C). Further quantitative analysis of this issue is really beyond the point, though, since we (and others, e.g., Eshleman et al. 2020) have already shown that 1,10-pt treatment induces a strong stress response that will lead to

TF and HAT promoter loss (as indeed shown for NuA4 (Epl1) at most genes, in our Figure 2), making it impossible to determine the causal relationship between transcription and promoter nucleosome acetylation using this method.

2. Manipulation of Y axes scales in Figure 2D to emphasize the difference in Epl1 binding is not appropriate and statistical analysis should be included.

As pointed out above we have now plotted the data with uniform y-axes and present boxplots of these data with statistical (p-value) analysis or their significance. We thank the reviewer for alerting us to this perceived subterfuge on our part, which was completely unintended.

Reviewer #5

In the rebuttal to the Matter Arising, Martin and Howe argue that careful analysis of the ChIP-seq data with strictly (and more appropriately) defined promoter boundaries show both increase and decrease in Epl1 binding upon 1,10 phenanthroline, while histone acetylation is mainly decreased in both cases. They therefore conclude that loss of acetylation cannot be simply explained by loss of HAT recruitment. Martin and Howe's response to this issue is convincing and highlights the importance of carefully defining genomic regions for data analysis.

We discuss extensively above why the Martin & Howe response is not convincing at all and why "the importance of carefully defining genomic regions for data analysis" does not explain our differences, since we used exactly the same region defined by Martin et al. (in their Figure 2d-f) to quantify Epl1 ChIP-seq signal. Furthermore, as pointed out above, the bizarre behavior of the small outlier group of genes at which Epl1 binding spikes dramatically following 1,10-pt treatment cannot be used to argue that promoter nucleosome acetylation is dependent upon transcription. In any event, in the following paragraph this reviewer clearly acknowledges that Epl1 ChIP-seq data cannot be used as a reliable read-out for NuA4 binding, as we point out above.

In summary, Albert and colleagues have raised very important points to question some key conclusions from a seminal study which addresses a fundamental knowledge-gap in gene regulation. I recommend that this exchange is published with minor revision of the Martin and Howe response to better address the question of causality.

We thank this reviewer for recognizing the validity of our arguments and the inadequacy of the response from Martin & Howe.